# Activation and signaling mechanism revealed by GPR119-G$_s$ complex structures

Yuxia Qian[1,5], Jiening Wang[2,5], Linlin Yang[3,5], Yanru Liu[1], Lina Wang[3], Wei Liu[1], Yun Lin[1], Hong Yang[2], Lixin Ma[2], Sheng Ye[1,4] ✉, Shan Wu[2] ✉ & Anna Qiao[1] ✉

Agonists selectively targeting cannabinoid receptor-like G-protein-coupled receptor (GPCR) GPR119 hold promise for treating metabolic disorders while avoiding unwanted side effects. Here we present the cryo-electron microscopy (cryo-EM) structures of the human GPR119-G$_s$ signaling complexes bound to AR231453 and MBX-2982, two representative agonists reported for GPR119. The structures reveal a one-amino acid shift of the conserved proline residue of TM5 that forms an outward bulge, opening up a hydrophobic cavity between TM4 and TM5 at the middle of the membrane for its endogenous ligands-monounsaturated lipid metabolites. In addition, we observed a salt bridge between ICL1 of GPR119 and Gβ$_s$. Disruption of the salt bridge eliminates the cAMP production of GPR119, indicating an important role of Gβ$_s$ in GPR119-mediated signaling. Our structures, together with mutagenesis studies, illustrate the conserved binding mode of the chemically different agonists, and provide insights into the conformational changes in receptor activation and G protein coupling.

GPR119 is a cannabinoid receptor-like class A G protein coupled receptor (GPCR), highly expressed in pancreatic β cells and intestinal enteroendocrine L cells, playing critical roles in glucose homeostasis and feeding behavior[1,2]. GPR119 elicits its physiological responses by coupling primarily to G$_s$ proteins to activate adenylate cyclase and cyclic AMP signaling[3]. Activation of GPR119 stimulates glucose-dependent insulin release from the pancreas and intestinal secretion of incretins, including glucagon-like peptide-1 (GLP-1) and glucose-dependent insulinotropic peptide (GIP)[4–6], also suppresses food intake in rats and reduces body weight gain[7]. Therefore, GPR119 becomes an attractive target for the development of novel therapeutics towards metabolic disorders, such as obesity and type 2 diabetes. In addition, targeting the GPR119/incretin axis had recently been indicated to be a promising new therapy for metabolic-associated fatty liver disease[8].

The endogenous agonists of GPR119 are endocannabinoid-like lipid metabolites[9,10], including N-oleoylethanolamine (OEA), 2-oleoylglycerol (2-OG), lysophosphatidylcholine (LPC), and *N*-Oleoyldopamine (OLDA)[2,7,11–14], the monounsaturated analogs of endogenous ligands of cannabinoid receptors (CB1) such as anandamide (AEA) and 2-arachidonoyl glycerol (2-AG) (Supplementary Fig 1). Despite their structural similarities, the endogenous ligands of GPR119 and CB1 selectively recognize their own receptors. For decades, the massive interest in GPR119 has resulted in numerous synthetic GPR119 agonists[15–17]. AR231453 and MBX-2982 are two representative ones among them. AR231453 is the first potent and orally efficacious agonist reported for GPR119[6]. AR231453 significantly increases cAMP accumulation and insulin release in both a hamster β-cell line and rodent islets, and improved glycemic control in normal and diabetic mice, however,

[1]Frontiers Science Center for Synthetic Biology (Ministry of Education), Tianjin Key Laboratory of Function and Application of Biological Macromolecular Structures, School of Life Sciences, Tianjin University, Tianjin, P. R. China. [2]State Key Laboratory of Biocatalysis and Enzyme Engineering, Hubei Collaborative Innovation Center for Green Transformation of Bio-Resources, Hubei Key Laboratory of Industrial Biotechnology, School of Life Sciences, Hubei University, Wuhan, Hubei, China. [3]Department of Pharmacology, School of Basic Medical Sciences, Zhengzhou University, Zhengzhou, China. [4]Life Sciences Institute, Zhejiang University, Hangzhou, Zhejiang, China. [5]These authors contributed equally: Yuxia Qian, Jiening Wang, Linlin Yang. ✉e-mail: sye@tju.edu.cn; wushan91@hubu.edu.cn; anna.qiao@tju.edu.cn

not in GPR119-deficient mice[18]. MBX-2982 shows positive results in phase II clinical trials of type 2 diabetes by successfully reducing post-prandial glucose levels in type 2 diabetes mellitus (T2DM) patients, and increases insulin and incretin levels in a 4-week phase II clinical trial (ClinicalTrials.gov Identifier: NCT01035879). How GPR119 recognizes endogenous and synthetic ligands and transduces signals remains a mystery. To provide molecular details of the binding and activation of GPR119, we present the cryo-electron microscopy (cryo-EM) structures of AR231453 or MBX-2982 activated full length GPR119 in complex with its down-stream heterotrimeric $G_s$ protein. The structures provide a snapshot into the agonist binding properties, its activation of GPR119, and the structural basis of G protein coupling. This work sets the framework to integrate a large body of structure-activity relationship (SAR) studies towards understanding GPR119 activation by different classes of ligands.

## Results

### Structure determination of GPR119-$G_s$ signaling complexes

We assembled GPR119-$G_s$ complexes by co-expression of the receptors with a dominant-negative $G\alpha_s$, and human $G\beta_1\gamma_2$[19]. The complex was formed on the membrane of insect cells in the presence of Nb35, which stabilizes the nucleotide-free complex by bridging the $G\alpha_s$ and $G\beta\gamma$ subunits. The structures of GPR119-Gs-Nb35 in complex with AR231453 or MBX-2982 were determined by single-particle cryo-EM with overall resolutions of 2.87 and 2.33 Å, respectively (Fig. 1a–d, Supplementary Fig. 2, and Supplementary Table 1).

Both cryo-EM maps of the AR231453- and MBX-2982-GPR119-Gs complexes exhibit well-resolved features for most amino acids, especially around the ligand-binding pocket, and clear density for AR231453, and MBX-2982 (Supplementary Fig. 2), allowing us to model the 7TM elements of GPR119, the $G_s$ heterotrimer, Nb35, and the two agonists. Both GPR119 structures share a similar conformation,

including a pronounced outward movement of TM6 at the cytoplasmic end relative to other class A GPCR structures in their inactive states, which is a hallmark of class A GPCR activation. The root mean square (r.m.s.) deviation for 279 Cα atoms, the majority of the receptor, is 0.57 Å. And both agonists, AR231453 and MBX-2982, adopt an extended conformation to bind in the similar binding pocket (Fig. 1), demonstrating the conservation of the activation mechanism of GPR119.

### Structural features of GPR119

The overall GPR119 structure shares a canonical architecture with previously solved class A GPCR structures, containing seven transmembrane α-helices (TM1-TM7) connected by three extracellular loops (ECL1-ECL3) and three intracellular loops (ICL1-ICL3) and an amphipathic helix (H8) (Fig. 1). However, three unusual aspects distinguish GPR119 from other class A GPCRs.

The first unusual aspect of the GPR119 structure is the conformation and orientation of ECL2, which consists of 23 residues (143–165) and folds into an intricate structure covering the binding pocket from the extracellular side (Fig. 2a). A highly conserved disulfide bond between C155 in ECL2 and C78[3.25] (Fig. 2a) (Ballesteros–Weinstein numbering used in superscript)[20] in helix III effectively ties ECL2 to the transmembrane core, and limits the extent of the conformational change of ECL2 during receptor activation. The distal portion of ELC2 makes close contacts with ECL1, further stabilizing ECL2. The remaining phenylalanine rich ECL2 after the conserved disulfide bond (156–165) snakes across, forming extensive hydrophobic interactions with TM4, TM6, and TM7.

Second, the GPR119 structure reveals a noncanonical consensus structural scaffold that is constituted by a network of non-covalent contacts between residues on the TM helices, in contrast to that of canonical class A GPCRs[21]. Among a consensus network of 24 inter-TM

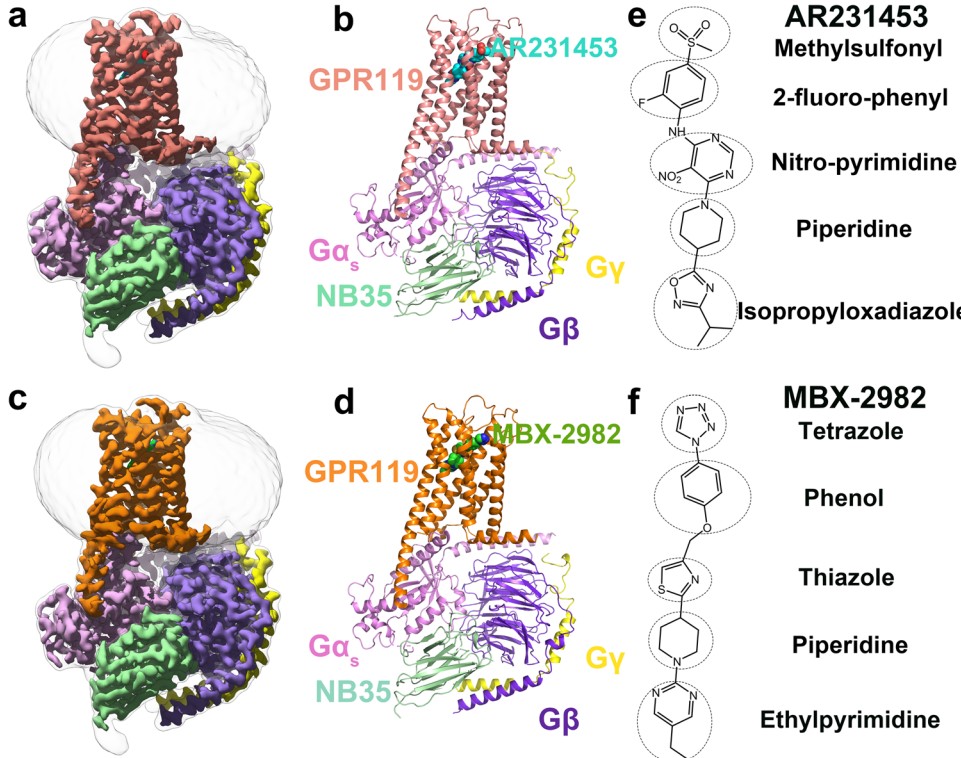

**Fig. 1 | Overall cryo-EM structures of the GPR119-$G_s$ heterotrimer complexes. a, b** The cryo-EM density (**a**) and the cartoon representation (**b**) of the AR231453-GPR119-$G_s$ complex. GPR119, AR231453, Gαs, Gβ, Gγ, Nb35 are colored pink, cyan, light purple, purple, yellow, light green, respectively. **c, d**, The cryo-EM density (**c**) and the cartoon representation (**d**) of the MBX-2982-GPR119-Gs complex. GPR119 in orange, MBX-2982 in green. **e, f** The chemical structures of AR231453 (**e**) and MBX-2982 (**f**).

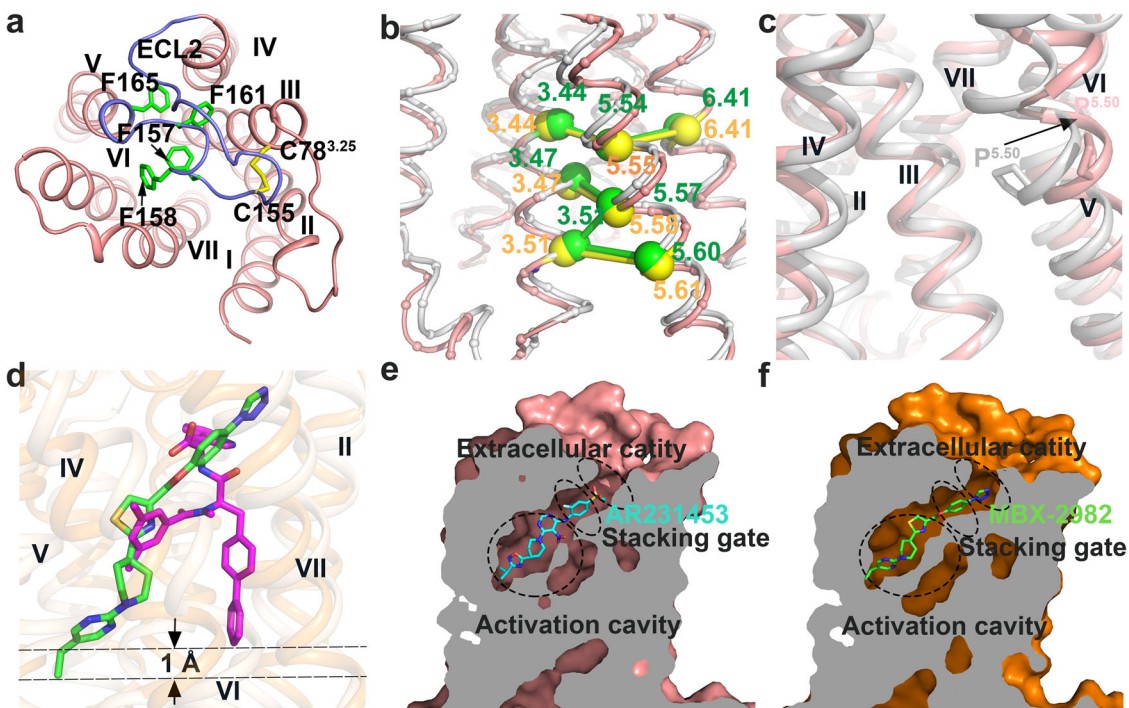

**Fig. 2 | Unique structure features and binding pocket. a** The second extracellular loop (ECL2) of GPR119 in an extracellular view. ECL2 is colored blue. The phenylalanines and disulfide bonds are shown as green and yellow sticks, respectively. **b** GPR119 displays a noncanonical consensus scaffold of non-covalent contacts in contrast to that of canonical class A GPCRs. GPR119 (pink, this study) and β2AR (gray, PDB ID: 3SN6) are superimposed with the Cαs of the 5 non-conserved inter-TM contacts shown as yellow and green spheres, linked by sticks, respectively. **c** The one-amino acid shift of the conserved Pro176$^{5.50}$ results in an outward bulge of TM5. GPR119 (pink, this study) and β2AR (gray, PDB ID: 3SN6) are superimposed

with Pro$^{5.50}$ shown as stick. The black arrow shows the one-amino acid shift of the conserved Pro$^{5.50}$. **d** Ligand binding pocket comparison between MBX-2982-GPR119 (this study) and IRL2500-ETBR (PDB ID:6k1q). GPR119, MBX-2982, ETBR, and IRL2500 are colored orange, green, light yellow, and magenta, respectively. **e, f** Cutaway view showing AR231453 (cyan) in the ligand binding pocket of GPR119 (pink) (**e**) and MBX-2982 (green) in that of GPR119 (orange) (**f**). The three functional compartments, an activation cavity, a stacking gate, and an extracellular cavity are indicated.

contacts in both active and inactive states mediated by 36 topologically equivalent amino acids[21], five of the contacts (3.44:5.54, 3.47:5.57, 3.51:5.57, 3.51:5.60, and 5.54:6.41, Ballesteros–Weinstein numbering used), all involving residues of TM5, are not conserved in GPR119 (Supplementary Fig. 3 and Supplementary Data 1). Such a discrepancy stems from the specific orientation of TM5 of GPR119. Structural comparison between GPR119 and β2-adrenergic receptor (β2AR)[22] revealed a one-amino acid shift in TM5, resulting in five new inter-TM contacts (3.44:5.55, 3.47:5.58, 3.51:5.58, 3.51:5.61, and 5.55:6.41) specific for GPR119 (Fig. 2b). As a consequence, the non-conservation of GPR119 structural scaffold should significantly reduce the accuracy of homology modeling for receptor-ligand interactions and drug discovery. Indeed, the modeled AR231453-GPR119 interactions[23,24] were significantly different to that experimentally observed in this study.

Third, a key feature of class A GPCR architecture is the presence of kinks in TM5, TM6 and TM7 caused by conserved proline residues. The one-amino acid shift of the conserved P176$^{5.50}$ that induces helical deformation, results in an outward bulge of TM5, which is significantly different to the inward bulge of TM5 observed in the canonical class A GPCRs, as shown in comparison to that of β2AR (Fig. 2c). Consequently, the **P**$^{5.50}$**I**$^{3.40}$**F**$^{6.44}$ motif observed in the canonical class A GPCRs is not present in GPR119. Moreover, the outward bulge of TM5 also opens up a hydrophobic cavity between TM4 and TM5 at the middle of the membrane, and significantly changes the ligand-binding pocket, as will be discussed later.

### An unusual ligand-binding pocket of GPR119
The noncanonical consensus structural scaffold of GPR119 defines an unusual ligand-binding pocket significantly different from the

previously observed ones from other class A GPCRs, such as the β2 adrenergic receptor[22], and the cannabinoid receptors[25–27].

First, both agonists penetrate deeply within the GPR119 pocket, approximately to the middle of the membrane, with the ligand penetration into the TM bundle deeper than those of all the lipid receptor (Supplementary Fig. 4), and all the class A GPCRs with structures thus far. Compared to IRL2500 in human endothelin ET$_B$ receptor[28], the deepest depth of orthosteric ligand penetration reported thus far, MBX-2982 penetrates within the pocket 1 Å deeper than IRL2500 in the pocket of human endothelin ET$_B$ receptor (Fig. 2d).

Second, the ligand-binding pocket of GPR119 is defined by all the seven TMs, as well as the ECL2 region. The prototypical GPR119 agonist AR231453 adopts an extended conformation, and binds aslant with an extreme tile-angle of ~45° in the ligand-binding pocket, in which the isopropyl moiety at one end buries deeply at the middle of the membrane, and the methylsulfonyl moiety at the other end is located at the extracellular side of the membrane (Fig. 2e).

Third, as might have been anticipated for a lipid-activated receptor, the ligand-binding pocket of GPR119 is highly hydrophobic. All 17 residues within 4 Å of AR231453 are hydrophobic. These hydrophobic residues line the ligand-binding pocket and make a variety of hydrophobic contact with the agonists. These residues are absolutely conserved across the vertebrate lineage (Supplementary Fig. 5).

### Agonist interactions in GPR119 ligand-binding pocket
The two representative synthetic agonists of GPR119 used in this study are chemically different. AR231453 contains a central pyrimidine core including a nitro-pyrimidine moiety and a piperidine, with a methylsulfonylphenyl and an isopropyloxadiazole termini (Fig. 1e). MBX-2982

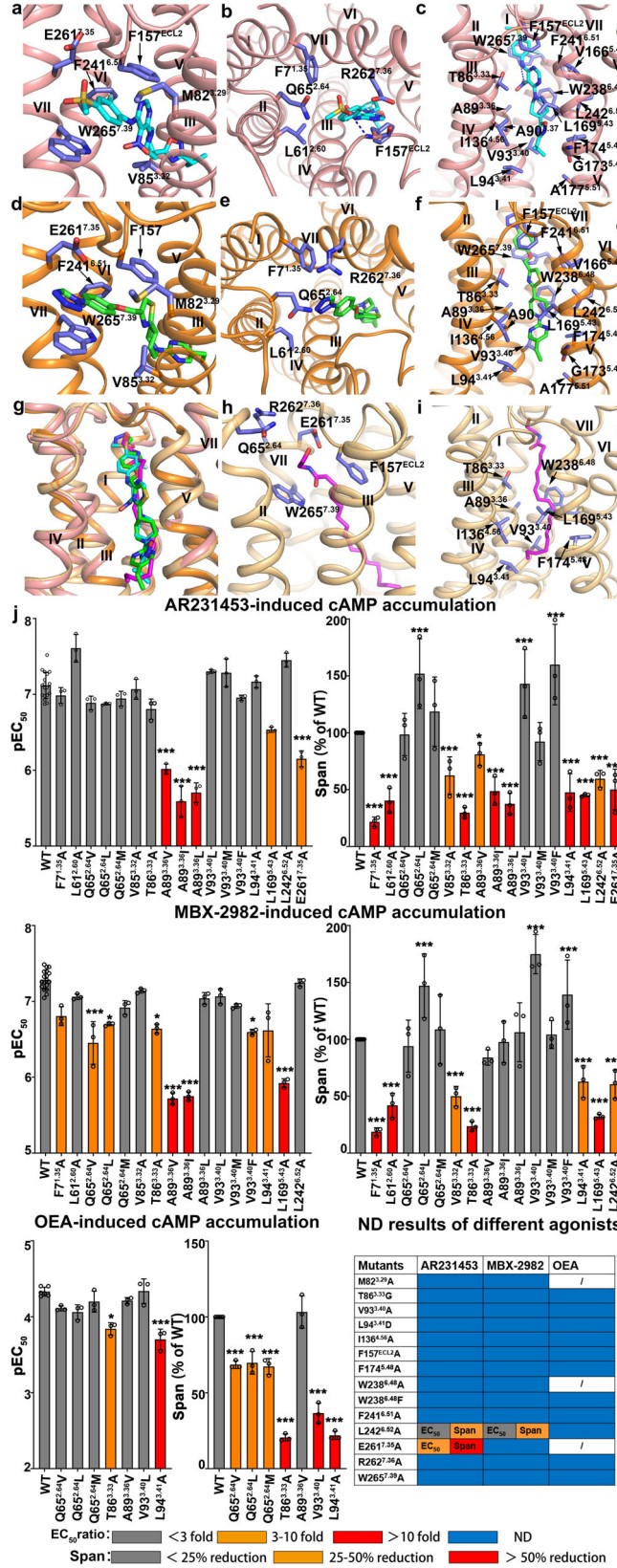

**Fig. 3 | Agonist interactions in GPR119 ligand-binding pocket. a–f** Ligand-receptor interactions in stacking gate (**a, d**) between GPR119 and AR231453 (**a**) or MBX-2982 (**d**), in extracellular cavity (**b, e**) between GPR119 and AR231453 (**b**) or MBX-2982 (**e**), and in activation cavity (**c, f**) between GPR119 and AR231453 (**c**) or MBX-2982 (**f**). **g** Superimposition of OEA with AR231453 and MBX-2982 in GPR119 ligand-binding pocket. **h, i** Ligand-receptor interactions in extracellular cavity and stacking gate between GPR119 and OEA of the docking model (**h**) and in activation cavity (**i**). **j** cAMP accumulation of GPR119 induced by agonist AR231453, MBX-2982 and OEA. (EC50 ratio = EC50 of mutant/EC50 of wild type) and maximal agonist response (span for cAMP accumulation) for each mutant relative to the wild-type receptor are shown according to the extent of effect. Data are from at least three independent experiments performed in technical triplicate. For AR231453/MBX-2982-induced cAMP accumulation, WT were repeated 18 times. For OEA-induced cAMP accumulation, WT were repeated 6 times. Other mutants were all repeated 3 times. $*P < 0.01$; $***P < 0.0001$ by one-way ANOVA followed by Dunnett's post-test, compared with the response of the WT. ND (not determined) refers to data where a robust concentration response curve could not be established within the concentration range tested. A detailed statistical evaluation is provided in Supplementary Tables 2 and 4. Source data are available as a Source Data file.

relationship (SAR) data of the discovery of AR231453[6], and the SAR data from the discovery of a series of novel thiazole derivatives starting from MBX-2982 as agonists for GPR119[29]. In addition, to correlate the structural observations with the ligand binding activity, we individually mutated most of the ligand pocket residues and assessed their effects on expression levels, and their ability to stimulate cyclic AMP (cAMP) production by either AR231453 or MBX-2982. We observed that the agonist interactions in GPR119 ligand-binding pocket are also in excellent agreement with the results from the functional and mutagenesis studies.

The ligand binding pocket can be divided into three functional compartments: a stacking gate mainly formed by two aromatic residues, an extracellular cavity close to the extracellular side of the membrane, and an activation cavity close to the toggle switch residue W238[6.48] (Fig. 2e, f).

The stacking gate is mainly formed by two aromatic residues, F157[ECL2] of ECL2, and W265[7.39] of TM7, with M82[3.29] and V85[3.32] at one side, and F241[6.51] and the aliphatic part of E261[7.35] at the other side, bridging the activation cavity and the extracellular cavity (Fig. 3a, d). The 2-fluoro-phenyl ring of AR231453, or the phenol ring of MBX-2982 stacks between F157[ECL2] and W265[7.39]. And the 2-fluoro group of AR231453 forms a halogen…π interaction with the benzene ring of F241[6.51] (Fig. 3a, d). Replacement of the 2-fluoro group by 3-fluoro group of AR231453 reduces the potency by 5-fold (Supplementary Figs. 6a, b and7a, b)[6], and addition of a 2-fluoro group at the phenol ring of MBX-2982 enhances the agonist potency by 4-fold (Supplementary Figs. 6i, j and 7i, j)[29], demonstrating the importance of stacking gate for receptor activation. The three aromatic residues and M82[3.29] play critical roles in GPR119 activation, as the alanine mutations of F157[ECL2], W265[7.39], F241[6.51], and M82[3.29] failed to activate GPR119 (Fig. 3j, Supplementary Fig. 8, and Supplementary Table 2). In addition, E261[7.35] is more important than V85[3.32], as the alanine mutation of E261[7.35] reduces the EC50 of AR231453 by 9-fold, and the cAMP production by more than half, and significantly reduces the effect of MBX-2982, whereas the alanine mutation of V85[3.32] has little effect on the agonist potencies of AR231457 and MBX-2982 (Fig. 3j, Supplementary Fig. 8, and Supplementary Table 2).

The extracellular cavity is formed by TM1, 2, 3, and 7, and ECL2. For AR231453, the sulfonyl group of the 4-methylsulfonyl moiety forms a hydrogen bond with backbone amide group of F157[ECL2], and the methyl group of the 4-methylsulfonyl moiety interacts with the side chains of L61[2.60] and M82[3.29] (Fig. 3b). The 4-methylsulfonyl moiety is specific and highly selective for GPR119, and its replacement by ethylsulfonyl or thioether moieties, reduces the potency by more than 10- or 100-fold (Supplementary Figs. 6a, c, d, and 7a, c, d)[6]. While for MBX-2982, the tetrazole moiety forms hydrophobic interactions with

contains five linearly linking moieties, ethylpyrimidine, piperidine, thiazole, phenol and tetrazole (Fig. 1f). However, despite the differences in chemical composition, both agonists assume a same overall extended geometry, with an overlapping ligand binding pocket (Fig. 2e, f). Moreover, the agonist interactions in GPR119 ligand-binding pocket are in excellent agreement with the structure activity

F7[1.35] and L61[2.60], and is close to two polar residues, Q65[2.64] and R262[7.36] (Fig. 3e). This explains the SAR data of the tetrazole moiety modified derivatives of MBX-2982, in which the replacement of the tetrazole moiety with pyrrolidine-2,5-dione, displayed a 2-fold enhanced agonistic effect in comparison with that with pyrrolidin-2-one, a carbonyl group absent form of pyrrolidine-2,5-dione (Supplementary Fig. 6j, k and 7j, k)[29]. The alanine mutation of R262[7.36] failed to activate GPR119. Three hydrophobic mutations of Q65[2.64] (Q65[2.64]L, Q65[2.64]Vm and Q65[2.64]M) all reduced the agonist potencies of MBX-2982 more severely than that of AR231453 (Fig. 3j, Supplementary Fig. 8, and Supplementary Table 2), which is reasonable given that MBX-2982 is closer to Q65[2.64]. Interestingly, the alanine mutations of F7[1.35] and L61[2.60] had little effects on the EC50 of AR231457 and MBX-2982, while reduced the maximum level of $G_s$-mediated cAMP production by over 50% (Fig. 3j, Supplementary Fig. 8, and Supplementary Table 2).

The activation cavity, so named since it is close to the toggle switch residue W238[6.48], is formed by TM3, 4, 5 and 6, and ECL2. The cavity is surrounded by five aromatic residues, F157[ECL2], F174[5.48], W238[6.48], F241[6.51] and W265[7.39], together with eleven hydrophobic residues, T86[3.33], A89[3.36], A90[3.37], V93[3.40], L94[3.41], I136[4.56], V166[5.40], L169[5.43], G173[5.47], A177[5.51] and L242[6.52], with F157[ECL2] and F241[6.51] involved in both activation cavity and stacking gate (Fig. 3c, f). Inside the cavity, the piperidine moiety of both agonists is well positioned by the extensive interactions at two ends, forms hydrophobic interactions with A89[3.36], L169[5.43], and L242[6.52], adopts an energy-unfavorable, boat-shape conformation, and stacks with W238[6.48] (Fig. 3c, f). In both agonist-bound GPR119 structures, the toggle switch residue W238[6.48] adopts a conformation with the aromatic ring parallel to the piperidine moiety, likely in response to the compression exerted from the agonist. Indeed, mutations of the toggle switch residue W238[6.48] to alanine or phenylalanine failed to activate GPR119 (Fig. 3j), confirming its importance in receptor activation. Mutations of A89[3.36] to other hydrophobic residues with larger side chain (Val and Ile) severely reduced the agonist potencies of AR231453 and MBX-2982 by more than 10-fold, and the alanine mutation of L169[5.43] also severely reduced the agonist potencies of AR231453 and MBX-2982 (Fig. 3j, Supplementary Fig. 8, and Supplementary Table 2), indicating that the relative position of the piperidine moiety is critical for receptor activation.

At one end of the activation cavity, the nitro-pyrimidine moiety of AR231453 stacks between T86[3.33] and W238[6.48], forming hydrophobic interactions with the side chains of F157[ECL2], L169[5.43] and F241[6.51], with the nitro group forming a hydrogen bond with the side chain amide group of W265[7.39] (Fig. 3c). The thiazole moiety of MBX-2982 overlaps with the nitro-pyrimidine moiety of AR231453, forming similar hydrophobic interactions while without any hydrogen bonds (Fig. 3f). Elimination of the side chain of T86[3.33] (T86G) failed to activate GPR119, while the alanine mutation of T86[3.33] (T86A) slightly reduces the agonist potency, demonstrating the importance of the stacking effect on agonist activation (Fig. 3j, Supplementary Fig. 8, and Supplementary Table 2). In addition, formation of a hydrogen bond likely stabilizes the relative position of AR231453 for receptor activation, as majority of the alanine mutations (T86A, L94A, I136A, L169A, L242A) reduce the potency of MBX-2982 (9-, 4-, 9-, 30-, and 29-fold) more severely than that of AR231453 (3-, 2-, 2-, 9-, and 9-fold) (Fig. 3j, Supplementary Fig. 8, and Supplementary Table 2). However, the effect of hydrophobic mutations of A89[3.36] is reversed (Fig. 3j, Supplementary Fig. 8, and Supplementary Table 2), given a steric hindrance effect on the nitro group of AR231453.

At the other end of the activation cavity, the isopropyloxadiazole moiety of AR231453 locates in a hydrophobic pocket close to mid-membrane, with the oxadiazole moiety surrounding by A90[3.37], V93[3.40], L169[5.43], and F174[5.48] (Fig. 3c). The isopropyl moiety of AR231453 forms close contacts with the side chains of L94[3.41], I136[4.56], and A177[5.51], and the Cα atom of G173[5.47] (Fig. 3c), perfectly explaining the SAR data of the alkyl substituent on the oxadiazole portion, in which increasing the size of this substituent from methyl to ethyl, and further to isopropyl

but not addition of more carbon atoms, provided a gradually enhanced improvement in agonist potency (Supplementary Figs. 6a, e, f and 7a, e, f)[6]. The ethylpyrimidine moiety of MBX-2982 overlaps with the isopropyloxadiazole moiety of AR231453, forming similar hydrophobic interactions (Fig. 3f). The ethylpyrimidine moiety is critical for MBX-2982, as its replacement by other moieties completely eliminated the agonistic activity (Supplementary Fig. 6h, l)[29]. In addition, similar to the SAR data of the discovery of AR231453[6], further elongation of the ethyl group of the ethylpyrimidine moiety also completely eliminated the agonistic activity (Supplementary Fig. 6h, m, n)[29]. Our docking study revealed that the three small molecules adopt unstable binding poses in the receptor (Supplementary Fig. 6l, m, n), which further explained the SAR data. Our mutagenesis study agreed perfectly with the SAR data. The alanine mutations of V93[3.40], F174[5.48] and I136[4.56] failed to activate GPR119 (Fig. 3j, Supplementary Fig. 8, Supplementary Table 2). Interestingly, replacement of V93[3.40], to other hydrophobic residues, such as leucine, methionine and phenylalanine, had little effect on the agonist potencies, indicating the importance of maintaining the hydrophobic interaction (Fig. 3j, Supplementary Fig. 8, and Supplementary Table 2). L169[5.43] is involved in the hydrophobic interactions in both ends and the piperidine moiety, and the alanine mutation of L169[5.43] severely reduced the agonist potency (Fig. 3j, Supplementary Fig. 8, and Supplementary Table 2).

## Molecular Docking of a representative GPR119 endogenous agonist

To further investigate the activation mechanism of endogenous agonists of GPR119, we performed docking of OEA. The top ranked docking pose of OEA assumes an overall extended geometry, with the hydrophilic head group locating at the extracellular cavity, and the long oleyl tail extending from the stacking gate to the activation cavity, very similar to that of AR231453 and MBX-2982 (Fig. 3g). This explains why OEA activates GPR119, instead of CB1 or CB2. Compared with the C-shaped conformation of the long aliphatic tail of AEA that contains four cis-double bonds, the oleyl tail of OEA only contains one cis-double bond and can adopt an extended conformation to fit into the binding pocket of GPR119 (Supplementary Fig. 1). We further performed functional and mutagenesis studies to verify the OEA binding mode. The model revealed that the hydrophilic head group of OEA interacted with E261[7.35] and stacked between F157[ECL2] and W265[7.39] (Fig. 3h). Correspondingly, the alanine mutations of F157[ECL2], E261[7.35] and W265[7.39] abolished GPR119-mediated cell signaling (Fig. 3j). The model also revealed that the oleyl tail stacks between T86[3.33] and W238[6.48] at one end, and locates in a hydrophobic pocket closed to mid-membrane at the other end, together to enable the cis-double bond of OEA directly stacking with W238[6.48] to activate the receptor. The hydrophobic interactions at both ends are important for OEA activation, as T86[3.33] and F241[6.51] at one end, and V93[3.40], L94[3.41], I136[4.56], L169[5.43] and F174[5.48] at the other end, are very sensitive to mutations (Fig. 3j). Interestingly, replacement of A89[3.36] to valine had little effect on the agonist potency of OEA (Fig. 3j), which is significantly different to those of AR231453 and MBX-2982 (Fig. 3j). Such a difference is explainable from the structures and the docking model, as AR231453 and MBX-2982 contain bulky ring structure close to A89[3.36]. Taken together, our mutagenesis and functional studies are in excellent agreement with the docking model, supporting an activation mechanism of OEA similar to that of AR231453 or MBX-2982, and providing insights into activation mechanism of other endogenous ligands with similar chemical structure.

## Structure of the GPR119·$G_s$ interface

The structure of the GPR119-$G_s$ complex reveals an overall similar mode of interactions, however, with notable structural differences, when compared to other $G_s$ bound receptors[22,30]. The C-terminal α5 helix of the Gα$_s$ subunit inserts into the cavity at the cytoplasmic site of

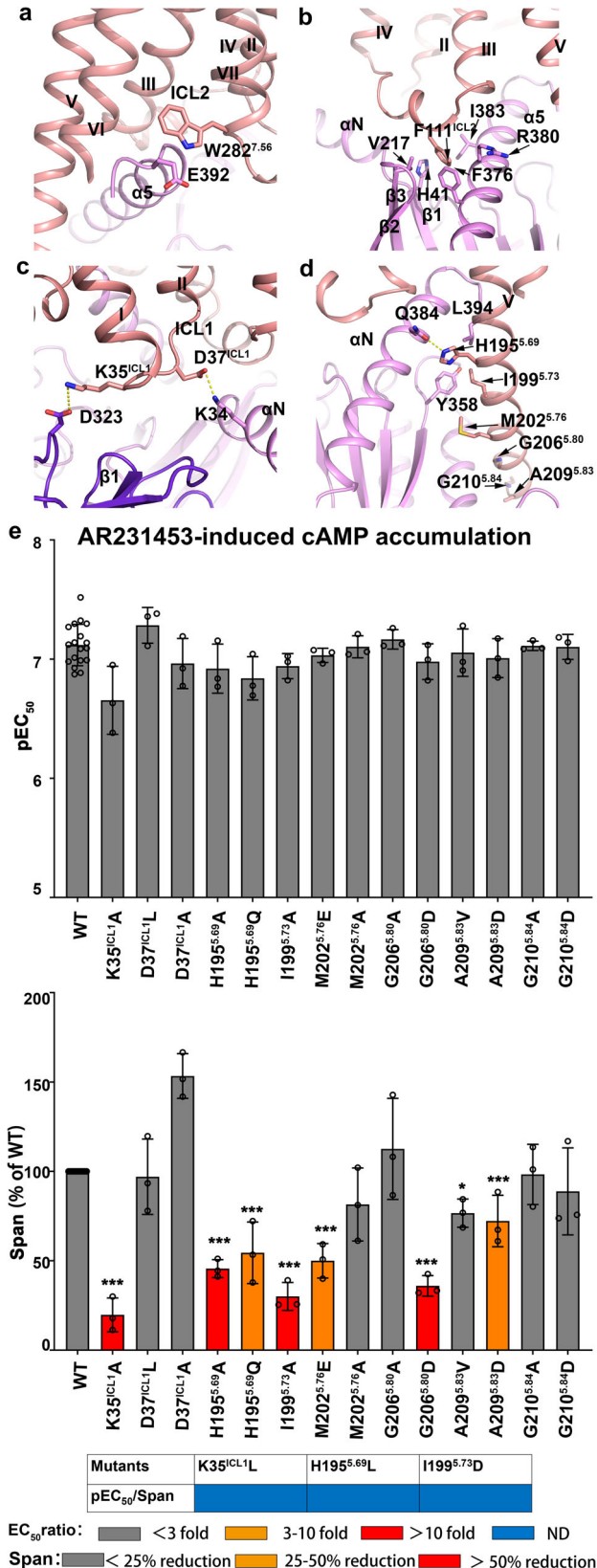

**e**

**AR231453-induced cAMP accumulation**

| Mutants | K35$^{ICL1}$L | H195$^{5.69}$L | I199$^{5.73}$D |
|---|---|---|---|
| pEC$_{50}$/Span | | | |

EC$_{50}$ratio: ▢ <3 fold  ▢ 3-10 fold  ▢ >10 fold  ▢ ND

Span: ▢ < 25% reduction  ▢ 25-50% reduction  ▢ > 50% reduction

**Fig. 4 | G$_s$ recognition pattern of GPR119. a** Interactions between the C-terminus of Gα α5 helix and GPR119. The π-π interaction between Gα$_s$ and GPR119 is shown. **b** F111$^{ICL2}$ of GPR119 docks into a hydrophobic pocket on the Gα$_s$ surface. **c** The salt bridge between ICL1 in GPR119 and Gα$_s$ or Gβ1. **d** Interactions between TM5 of GPR119 and Gα$_s$. **e** cAMP accumulation of GPR119 induced by agonist AR231453. WT were repeated 18 times. Other mutants were all repeated 3 times. *P < 0.01; ***P < 0.0001 by one-way ANOVA followed by Dunnett's post-test, compared with the response of the WT. A detailed statistical evaluation is provided in Supplementary table 2. Source data are available as a Source Data file.

dopamine receptor-G$_s$ complex (PDB ID: 7JV5)[30]. In addition, the GPR119-G$_s$ interface involves ICL1, ICL2 and TM5. First, F111$^{ICL2}$ is buried in a hydrophobic pocket formed by the α5 helix, the αN-β1 loop and the β2-β3 loop of Gα$_s$, forming strong interactions, which are conserved in other class A GPCR-Gα$_s$ complexes (Fig. 4b). The corresponding residue in β2AR is F139$^{ICL2}$. Alanine substitution of F139$^{ICL2}$ in β2AR and other G$_s$ coupling receptors impairs their ability to activate G proteins[31]. Second, the cytoplasmic end of TM5 of GPR119 is extended by two additional helical turns compared to that of β2AR, which forms extra interactions with the Ras-like domain of Gα$_s$ in the assembly of the receptor-G$_s$ complex (Fig. 4d). Mutagenesis studies revealed that all mutations of H195$^{5.69}$ and I199$^{5.73}$, either to hydrophobic or hydrophilic residues, and the point mutants of M202$^{5.76}$, G206$^{5.80}$ and A209$^{5.83}$, designed to disrupt the hydrophobic interface by changing hydrophobic residues to acidic residues, significantly reduced the maximum level of G$_s$-mediated cAMP production (Fig. 4e), indicating that the hydrogen bond between H195$^{5.69}$ and Q384$^{G.H5.16}$, and the hydrophobic interactions mediated by I199$^{5.73}$ (with Y358$^{G.h4s6.20}$ and L394$^{G.H5.26}$) are more important. Third, ICL1 forms two salt-bridges, one with Gα$_s$ (D37$^{ICL1}$-K34$^{G.HN.51}$), and the other one with Gβ$_s$ (K35$^{ICL1}$-D323) (Fig. 4c). Disruption of the salt bridge with Gβ$_s$ (K35L or K35L) either completely eliminated or significantly reduced the agonistic activity (Fig. 4e), indicating Gβ$_s$ is critical to G$_s$ protein coupling to GPR119. While disruption of the salt bridge with Gα$_s$ (D37L) had little effect on the agonist potency (Fig. 4e). Taken together, our structural and functional data suggested that the interaction between ICL2 and Gα$_s$, and that between ICL1 and Gβ are both important, such that G$_s$ can adopt a proper orientation to efficiently couple with GPR119.

## Discussion

Delineating the GPR119 structural basis for ligand recognition and G protein recruitment will facilitate rational design and development of drugs with high affinity and selectivity as well as optimal therapeutic effects. Here we report the cryo-EM structures of two agonist-bound GPR119-Gs signaling complex and detailed interactions between the agonists and the receptor. The structures reveal structural insights into GPR119 activation. In both structures, the agonist directly interacts with the toggle switch residue W238$^{6.48}$ with the piperidine moiety adopting an energy-unfavorable boat-shape conformation, and stacking with W238$^{6.48}$ (Fig. 3e, f). The observation leads us to propose that the relative position of the piperidine moiety is critical, for either receptor activation or inactivation. The piperidine moiety is well positioned by the extensive interactions at two ends of the activation cavity, and adopting an energy-unfavorable boat-shape conformation. Indeed, our mutagenesis data reveal that the majority of the hydrophobic interactions in the activation cavity are significantly important for GPR119 activation by agonists. Moreover, a derivative of AR231453 that shares a central pyrimidine core containing a nitro-pyrimidine moiety and a piperidine moiety (Supplementary Fig. 9a), while without the isopropyloxadiazole moiety, is an inverse agonist[6]. This suggests that its piperidine moiety most likely locates in a similar, while different relative position. Correspondingly, the piperidine moiety of the inverse agonist changes the conformation of toggle switch residue W238$^{6.48}$ to inactivate the receptor. During the last two decades, a whole spectrum of synthetic GPR119 agonists were reported, including

GPR119 to form the major interaction interface with residues from TM3, ICL2, TM5, TM6 and TM7 of the receptor (Fig. 4a). Notably, E392 at the C-terminus of α5 helix of the Gα$_s$ subunit forms a stacking interaction with W282$^{7.56}$ at TM7 of GPR119, which was not observed in the structures of other class A GPCR-G$_s$ complex, including β2-adrenergic receptor (β2AR)-G$_s$ complex (PDB ID: 3SN6)[22] and D1

the relatively short ones with different chemical scaffold comparing to those of AR231453 and MBX-2982, such as AS1269574 (Supplementary Fig. 9b)[32]. Our study suggests that they might locate in the activation cavity and directly interact with the toggle switch residue W238[6.48], however, awaiting further experimental validation.

Currently, no structure of the inactive GPR119 is available to allow proper structural comparison with the active GPR119. Nevertheless, the GPR119 structure (UniProt ID: Q8TDV5) was recently predicted by AlphaFold 2.0 (https://alphafold.ebi.ac.uk/)[33] in a ligand-free condition without any interacting proteins, such as G proteins. The overall structure is very similar to the agonist-bound GPR119 structures determined in this study, with r.m.s. deviations of 2.0-2.1 Å for 279 Cα atoms of the receptor. The largest difference between the predicted and the experimentally determined GPR119 structures is a 12 Å outward movement of TM6 when measured at the Cα carbon of Asp220 (Supplementary Fig. 10a), very similar to that observed between inactive and active CB1s[25–27,34,35], β2 adrenergic receptors[22,36], rhodopsins[37,38], M2 muscarinic receptors[39,40], μ-opioid receptors[41,42] and A2A adenosine receptors[43,44], indicating that it represents an inactive state.

The agonist-bound GPR119 structures, together with the predicted ligand-free one, revealed an interesting aspect that many of the amino acids that interact with agonists are not optimally positioned for agonist binding in the predicted GPR119 structure. That is, there is no preformed binding site for the agonist. Compared with the AR231453-bound GPR119 structure, TM3, TM4 and ECL2 of the predicted GPR119 structure are downward by approximately 1.8, 1.0 and 1.9 Å, while TM2 is upward by ~0.8 Å, and TM1 is inward by approximately ~0.8 Å, relative to TM5 (Supplementary Fig. 10b). This effectively distorts the ligand-binding site, resulting in a network of noncovalent interactions between side chains of the residues that interact with agonists, thus stabilizing the inactive state of the receptor. These interactions include a hydrophobic contact between A89[3.36] and the toggle switch residue W238[6.48], and that between F157[ECL2] and W265[7.39] (Supplementary Fig. 10c). And thus, for an agonist to bind, the intramolecular interactions that keep the receptor in an inactive state must be first broken. The ligand binding pocket undergoes an extensive structural rearrangement. The agonist acts as an interacting partner with these residues, serving as bridges that stabilize new interactions between TMs. In doing so, the agonist moves specific TMs closer to each other, pushes some of them further apart, or rotates one relative to the other.

To investigate the effects of agonist binding on GPR119 conformation, molecular dynamics (MD) simulations were performed on three systems, GPR119/AR231453 system (hereafter referred to as GPR119/AR), GPR119/MBX-2982 system (hereafter referred to as GPR119/MBX), and predicted ligand-free GPR119 system (hereafter referred to as AlphaFold). The r.m.s. deviations of GPR119 residues and bound ligands showed that they reached stability during the simulation (Supplementary Fig. 11a, Supplementary Fig. 12a, d). GPR119 was more dynamic in agonist-bound systems than that without agonist predicted by AlphaFold 2.0 (Supplementary Fig. 11a). But their global conformations were similar to the initial conformation in each system (Supplementary Fig. 11b). Especially, at the intracellular end, TM6 in the two agonist-bound systems maintained the active conformation with larger TM3-TM6 distance than that predicted by AlphaFold 2.0 (Supplementary Fig. 11c, d). These results suggested that agonists could help stabilize GPR119 in active state to certain extent in the absence of G protein. As for ligand binding, AR231453 adopted more stable binding poses relative to MBX-2982 due to more polar interactions with GPR119 (Supplementary Fig. 12c, f). Accordingly, residues around AR231453 displayed the same movements in three simulations when compared to those in the ligand-free system (Supplementary Fig. 12b). The observed structural rearrangement in MD simulations were contributed by W238[6.48], F157[ECL2] and W265[7.39], highly consistent

with the comparison between static structures (Supplementary Fig. 10c). While residues around MBX-2982 displayed more flexibility during independent simulations (Supplementary Fig. 12e), which was in line with the less stable binding of MBX-2982.

In this study, we report two relatively high resolution cryo-EM structures of GPR119 signaling complex bound to two representative agonists AR231453 and MBX-2982. The structures reveal a non-canonical consensus structural scaffold and an unusual ligand-binding pocket underlying ligand selectivity in contrast to those of canonical class A GPCRs. The structures, together with mutagenesis studies, also illustrate the conserved binding mode of the chemically different agonists and the conformational changes in receptor activation, and provide insights into the G protein coupling. These results compose the structural framework to understand receptor modulation by agonists, and will assist rational approaches to therapeutic targeting of this receptor for metabolic disorders.

## Methods

### Constructs
The human GPR119 was modified to contain a hemagglutinin (HA) signal peptide and a thermally stabilized bRIL at N terminus, a Flag tag and a strep tag at C terminus. A single mutation (S237C) was to improve protein yield. A dominant negative Gα$_s$ (DNGα$_s$) construct was generated by site-directed mutagenesis to incorporate eight mutations including S54N, G226A, E268A, N271K, K274D, R280K, T284D, and I285T[19].

### Insect cell expression
Human GPR119, DNGα$_s$, His6-tagged human Gβ1 and Gγ2 were co-expressed in HighFive insect cells (Invitrogen) using the Bac-to-Bac Baculovirus Expression System (Invitrogen). Cell cultures were grown to a density of 2–3 million cells per ml and then infected with high-titer-viral stocks at a MOI (multiplicity of infection) ratio of 1:1:1 for GPR119, DNGαs, and Gβ1γ2. Cells were collected by centrifugation 48 h after infection and stored at −80 °C until use.

### Purification of AR231453- or MBX-2982-GPR119-Gs complexes
Cells were suspended in a buffer including 20 mM HEPES, pH 7.4, 50 mM NaCl, 2 mM MgCl$_2$ supplemented with protease inhibitor cocktail tablets (Roche). GPR119-G$_s$ complex was obtained by adding 5 μM AR231453 (MCE) or 20 μM MBX-2982 (MCE), 10 μg ml$^{-1}$ Nb35 (prepared as previously described) and 25 mU ml$^{-1}$ Apyrase; followed by 1 h incubation at 20 °C. Insoluble material was removed by centrifugation at 30,000×g for 30 min. The complex protein was solubilized in 25 mM HEPES, pH 7.4, 150 mM NaCl, 0.5% (w/v) lauryl maltose neopentyl glycol (LMNG, Anatrace), 0.025% cholesterol hemisuccinate (CHS, Anatrace), 2 mM MgCl$_2$, 25 mU ml$^{-1}$ Apyrase, 5 μM AR231453 or 20 μM MBX-2982 at 4 °C for 2 h. The supernatant was isolated by centrifugation and was further incubated with Strep-Tactin® XT (IBA) resin overnight at 4 °C.

The resin was washed with 20 column volumes of 25 mM HEPES, pH 7.4, 150 mM NaCl, 0.01% (w/v) LMNG, 0.0005% CHS, 2 mM MgCl$_2$, 5 μM AR231453 or 20 μM MBX-2982. Then the resin was eluted with five column volumes of 150 mM Tris-HCl, pH 8.0, 150 mM NaCl, 0.01% (w/v) LMNG, 0.0005% CHS (Anatrace), 2 mM MgCl$_2$, 50 mM biotin, 5 μM AR231453 (MCE) or 20 μM MBX-2982 (MCE). The complex protein was then purified by size-exclusion chromatography using a Superdex 200 Increase 10/300 column (GE Healthcare) pre-equilibrated with 20 mM HEPES, pH 7.4, 100 mM NaCl, 0.01% (w/v) LMNG, 0.0005% CHS, 2 mM MgCl$_2$, 5 μM AR231453, or 20 μM MBX-2982.

### Cryo-EM sample preparation and data collection
For AR231453- or MBX-2982-GPR119-Gs complexes, the cryo-EM grids were prepared by applying 3 μl aliquot sample at a concentration of 2 mg ml$^{-1}$ and 4.1 mg ml$^{-1}$ respectively to glow-discharged Quantifoil holey carbon grids (Au, R1.2/1.3, 300 mesh). The grids were then

blotted for 3 s and plunged into liquid ethane using a FEI Vitrobot Mark IV (Thermo Fisher Scientific) operated at 4 °C and 100% humidity and subsequently stored in liquid nitrogen for data collection.

Cryo-EM movie stacks were collected with a Titan Krios microscope (Thermo Fisher Scientific) at 300 kV equipped with a K3 summit electron direct detector (Gatan) in super-resolution mode at a nominal magnification of 105,000×, yielding a pixel size of 0.4255 Å. And a Gatan BioQuantum energy filter was operated at a slit width of 20 eV. Automated data collection software EPU was used for data collection with the defocus range varying from −1.0 to −1.5 μm. Each micrograph was dose-fractionated to 40 frames with a total exposure time of 2.5 s, resulting in a total dose of 54 electron per $Å^2$.

### Image processing
Original movie stacks were summed and corrected for drift and beam-induced motion by using MotionCor2[45] with a binning factor of 2. The contrast transfer function (CTF) parameters of each micrograph were estimated by Gctf[46]. Initial Model was created by cryoSPARC[47]. All manual and automatic particle picking, two-dimensional classification, three-dimensional classification and auto-refine as well as CTF refinement and polishing were performed with RELION-3.1.1[48].

For AR231453-GPR119-Gs complex, 2,229,562 raw particles that were autopicked and extracted from 3,159 micrographs with a pixel size of 1.702 Å were subjected to reference-free two-dimensional classification, which were used to discard bad particles. 2,220,441 particles displayed clear features after two rounds of two-dimensional averages were selected and split into four classes for three-dimensional classification. One good class showed detailed structural features containing 1,164,181 particles were subjected to 3D refinement, CTF refinement and Polishing using a pixel size of 0.851 Å. The final density map was post-processed in RELION, resulting in a map at 2.87 Å resolution.

For MBX-2982-GPR119-Gs complex, 3,985,018 raw particles that were autopicked and extracted from 4729 micrographs with a pixel size of 1.702 Å were subjected to reference-free two-dimensional classification, which were used to discard bad particles. 3,944,511 particles displayed clear features after two-dimensional averages were selected for two rounds of three-dimensional classification. A well-defined subset with 705,428 particles were subjected to 3D refinement, CTF refinement and Polishing using a pixel size of 0.851 Å. The final density map was post-processed in RELION with a global resolution of 2.33 Å. Local resolution map was estimated using RELION. All 3D density maps were displayed using UCSF Chimera[49].

### Model building and refinement
The initial template of GPR119 was generated by Modeller[50]. Meanwhile the atomic model of $G_s$ and Nb35 from the structure of the human glucagon receptor in complex with $G_s$ (PDB 6LMK)[51] were used as initial template of the $G_s$-Nb35 complex. Models of GPR119 and $G_s$-Nb35 were rigid-body fitted into electron microscopy density map using Chimera, followed by manual adjustment and rebuilding in COOT[52]. The coordinates and geometry restraints of AR231453 and MBX-2982 were generated in PHENIX[53] using phenix.elbow. All the models were further refined by real-space refinement in PHENIX[53]. The refinement statistics of final models were provided by MolProbity[54] and summarized in Extended Table 1. Structural figures were created by Chimera[49] and PYMOL (https://pymol.org/2/).

### Flow cytometry
The cell surface GPR119 expression level was detected by incubating 10 μl of cells with 10 μl of monoclonal anti-FLAG M2−fluorescein isothiocyanate antibody (Sigma-Aldrich) at 4 °C for 20 min in the dark. The fluorescent signal of the bound antibody was measured using a FACSCalibur (Becton Dickinson, Sunnyvale, CA). Single parameter histograms can be used to further identify distinct cell types that antibody-specific population of cells. Cells expressed GPR119 are gated according to negative cells without fluorescein isothiocyanate.

### cAMP assay
HEK293 cells (Invitrogen) were harvested 48 h after transfection with 1 μg ml$^{-1}$ plasmid. cAMP accumulation was measured using a HTRF cAMP kit (Cisbio Bioassays, 62AM6PEC) according to manufacturer's instructions. In brief, the HEK293 cells expressing GPR119 were seeded onto 384-well plates (5 μl, 4000 cells per well) and incubated at room temperature for 30 min with different concentrations of AR231453 ($10^{-5}$ M-$10^{-11}$M), MBX-2982 ($10^{-5}$ M-$10^{-11}$M) or OEA ($10^{-3}$ M-$1.28 \times 10^{-8}$ M). Then 5 μl detection reagent d2-conjugated cAMP and 5 μl cryptate (Eu)-conjugated antibody were added in each well. After incubation at room temperature for 1 h, the plates were read using a microplate reader (PerkinElmer) with excitation at 330 nm and emission at 620 nm and 665 nm. cAMP accumulation was analyzed by a standard dose-response curve using GraphPad Prism 7.0 (GraphPad Software). EC50 and pEC50 ± SEM were calculated using nonlinear regression (curve fit).

### Protein-ligand docking
To investigate the interaction modes of OEA and different derivatives of AR231453 and MBX-2982 in GPR119, a docking study was performed using AutoDock Tools package (version 1.5.6) and AutoDock Vina (version 1.1.2)[55,56]. In docking simulation, the endogenous ligand OEA and the derivatives of AR231453 and MBX-2982 were used as ligands and the cryo-EM structures of AR231453-GPR119 or MBX-2982-GPR119 were used as receptors. The receptors and ligand were respectively optimized and prepared to the pdbqt format files needed for docking. Docking grids were generated using enclosing boxes centered on MBX-2982 or AR231453 in the two cryo-EM structures. The processed ligands were then docked into two receptors, outputting the top 20 conformations of each docking run. Considering the effects of receptor flexibility on ligand binding, we conducted both rigid and flexible dockings for derivatives. Other parameters were set to the default. As OEA has a smaller volume and exhibits more conformational flexibility, rigid docking results were good enough. But for the derivatives, only flexible docking could produce reasonable binding poses for all ligands. The most reliable binding poses were then selected according to the favorable interaction energy and our visual inspection.

### Molecular dynamics simulations
Based on the cryo-EM structures, we first built complex models for systems GPR119/AR and GPR119/MBX by adding missing residues R213-S219 and back mutating the S237C mutation to serine. Together with the predicted AlphaFold structure, all models were separately placed into a 90 Å × 90 Å palmitoyl oleoyl phosphatidyl choline (POPC) bilayer generated by CHARMM-GUI web server (https://charmm-gui.org/).These systems were solvated in a box (90 Å × 90 Å × 110 Å) with TIP3P water molecules and 0.15 M NaCl. Each system was replicated to perform three independent simulations and each simulation was run up to 1000 ns.

MD simulations were carried out with GROMACS 2020 package[57] with isothermal−isobaric (NPT) ensemble and periodic boundary condition. The CHARMM36-CMAP force field[58] was applied for protein, POPC phospholipids, ions and water molecules. Ligand parameters were adapted from the CHARMM Generalized Force Field (CGenFF)[59,60]. Energy minimizations were first performed to relieve unfavorable contacts in the system, followed by three independent 50-ns equilibration simulations to relax each system with stepwise restraints on different components. Subsequently, a 1000 ns production run was performed for each simulation. SETTLE constraints[61] and LINCS constraints[62] were applied on the hydrogen-involved covalent bonds in water molecules and in other molecules, respectively, and the time step was set to 2 fs. Electrostatic interactions were calculated with

the Particle-Mesh Ewald (PME) algorithm[63] with a real-space cutoff of 1.0 nm. The temperature was maintained at 310 K using the v-rescale method[64] and the pressure was kept constant at 1 bar by semi-isotropic coupling to a Parrinello-Rahman barostat[65] with $\tau_p = 2.5$ ps and a compressibility of $4.5 \times 10^{-5}$ bar. Analysis of simulation data was conducted using PyMOL (http://www.pymol.org), tools implemented in GROMACS, and in-house scripts.

## Reporting summary

Further information on research design is available in the Nature Portfolio Reporting Summary linked to this article.

## Data availability

The data that support this study are available from the corresponding authors upon reasonable request. The cryo-EM density maps have been deposited in the Electron Microscopy Data Bank (EMDB) under accession codes EMD-32425 (GPR119-AR-Gs-Nb35) and EMD-32424 (GPR119-MBX-Gs-Nb35). The coordinates have been in the RCSB Protein Data Bank (PDB) under accession codes 7WCN (GPR119-AR-Gs-Nb35) and 7WCM (GPR119-MBX-Gs-Nb35). Source data underling Figs. 3 and 4 are available as a Source Data file. Source data are provided with this paper.

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

## Acknowledgements

This work was supported in part by Ministry of Science and Technology (2020YFA0908500 to S.Y. and 2020YFA0908400 to S.W.), the National Natural Science Foundation of China (31971127 to S.Y., 32000851 to A.Q., 31900895 to L.Y. and 31900930 to S.W), China Postdoctoral Science Foundation (2020M672434 to S.W.), and the Fundamental Research Funds for the Central Universities (to S.Y.).

## Author contributions

Y.Q. prepared the protein samples for cryo-EM and performed signaling assays with the assistance from Y.Liu and Y.Lin. J.W. performed cryo-EM sample preparation, acquired cryo-EM data, data processing and analysis. L.Y. and L.W oversaw the molecular docking and performed the molecular dynamic simulation. H.Y and L.M helped with cryo-EM data collection. W.L. assisted with cryo-EM data analysis. S.W. helped with cryo-EM data collection, analysis, and processing. S.Y. and A.Q. initiated the project, planned and analyzed experiments, supervised the research, and wrote the manuscript with input from all co-authors.

## Competing interests

The authors declare no competing interests.
