## [Peer Review File · Nature Communications]

Activation and Signaling Mechanism Revealed by GPR119-Gs Complex StructuresEditorial Note: This manuscript has been previously reviewed at another journal that is not operating a transparent peer review scheme. This document only contains reviewer comments and rebuttal letters for versions considered at Nature Communications.

Reviewers' Comments:

Reviewer #1:

Remarks to the Author:

In my initial review (for NSMB), I requested that the authors try and get the unliganded or apo state structure for GPR119. I understand that it has been technically challenging to get it and I have nothing more to ask for experiments. I support publication of the paper in its current form, with some language formatting.

The quality of the structures seems very good, and overall it adds to our knowledge of this orphan GPCR.

Minor points:

Page 5 line 8: synthetic, not synthesis.

page 7 line 6: extent, not extend.

page 8, line 10: compared, not comparison.

Reviewer #2:

Remarks to the Author:

In response to the questions raised, the authors basically solved my doubts. The description of the content of manuscript has been modified and the format of figure has been optimized, which make the text more logical and clear. However, some words in the manuscript are not precise. For example, Page 6, line 5, "for all amino acids": it is better to use "for most atoms acids".

Page 8, line 21, "with the deepest depth... thus far": it will be more appropriate to use "deeper...than" for these descriptions.

Other similar parts should be checked if I didn't point out.

Reviewer #3:

Remarks to the Author:

Reply from Reviewer 3

The manuscript has been improved with respect to a biological context of GPR119. However, the authors may consider to update the view here with the more recent literature, for instance doi.org/10.1186/s11658-021-00276-7.

The author could also improve the manuscript by providing overview of the two selected compounds and their path as putative lead compounds for GPR119

Moreover, the authors may want to correct a few language/typographic mistakes, for instance:

-page 5: line 5, line 8 and 9.

-page 11: line 2

-page 15: line 16

and other places.

Reviewer #4:

Remarks to the Author:

Qian et al. provide structural context for the active state of human GPR119 in the presence of two synthetic agonists and the receptor's cognate G protein. The authors present two high quality cryo-electron microscopy structures of the receptor and identify structural features that have not been reported for other class A G protein-coupled receptors, including an outward bulge in transmembrane helix 5 and a salt-bridge between intracellular loop 1 and the $G\beta$ subunit. Other protein-ligand interactions that could be relevant for GPR119 activation and signaling are identified from the structures and then experimentally verified through multiple point mutations. The authors also explore the binding mode of different agonists and their derivatives using a docking study and perform molecular dynamics simulations where they compare the dynamics of the ligand-free vs. the agonist-bound receptor. Overall, I believe that the study is thorough and that the reported findings will be of great interest for the G protein-coupled receptor community. However, the present manuscript would benefit from some revisions, particularly pertaining to the following points:

Observations related to the computational work

1. Page 21, line 14-16: It needs to be clarified what subset of atoms was included in the RMSD calculations that are reported in Figure 12a. The loops of GPCRs tend to be quite flexible and it is customary to exclude them when checking the stability of the system. Most groups report the RMSD of the $C\alpha$ s in the transmembrane domain. It is not clear from the text or the figure if this is what the authors did or not?
2. Page 32, line 24: The authors should provide a more complete description of how the initial structure of GPR119 used for refinement was generated. What templates did they use? What were their percentages of identity vs. GPR119? How many models did they generate with MODELLER and how were they ranked (e.g., based on their DOPE score)?
3. Page 34, line 5: Can the authors elaborate on the need to use rigid docking for OEA, but not for the other derivatives that were docked?
4. Page 34, line 5: Please provide the docking scores obtained from the docking studies.
5. Page 34, line 20: Do the authors mean "adapted" or "adopted"? Were there penalties higher than 10 in the parameters generated from CGenFF? If so, were the parameters optimized as suggested by Vanommeslaeghe et al. (2012) in their J. Chem. Inf Model 52(12), 3144-54 paper?
6. SI Page 12, line 4: How were the average/representative structures presented in Supplementary Figures 11b, 11d, 12c, 12e and 12f generated? Since it is not really meaningful to average coordinates, usually one would cluster the conformations sampled in the simulations and use the medoid from the most populated cluster as a representative structure.

Other observations

A. Missing references

1. Page 7, line 5: The reference to the Ballesteros-Weinstein paper needs to be added.

B. Data presentation:

1. Page 6, line 12-14: Figure 1 does not really show the two ligands overlapping, but in different panels. And would the fact that they overlap demonstrate that the activation mechanism is conserved?

I think that this sentence needs to be reworded to reflect what it is actually in the figure.

2. Figure 2e and f: The different cavities of the ligand binding pocket need to be more clearly defined, because their limits are not clear just based on the arrows shown in the panels. Perhaps it would be helpful if they were indicated with boxes instead?
3. Figure 3a-i: Adding BW-numberings to the residues would be helpful for the reader and it would also match the notation used in the bar plots.
4. Figure 4: Since this figure includes mutations in both the receptor and the G protein, I would suggest adding a prefix so that they are easier to identify (e.g., R_ for receptor residues, G α _ for G α residues and so on).
5. Supplementary Table 1: The value of the rotamer outliers for GPR119-MBX-Gs-Nb35 needs to be centered.

C. Technical clarifications:

1. Page 6, line 11 and page 20, line 7: What does the subset of 279 C α s correspond to? The transmembrane helices? It needs to be specified.
2. Page 12, line 17: How can ECL2 be part of the activation cavity? Is it not part of the extracellular cavity? Please see my suggestion above of presenting the different cavities in a clearer manner in Figure 2e and f.
3. Page 16, line 7-9: I was not able to find the results of the E261A mutant in Figure 3l.

D. Grammar/Phrasing:

1. I would suggest that the authors choose a different title, as the current one is practically identical to Hua et al.'s (2020): "Activation and Signaling Mechanism Revealed by Cannabinoid Receptor-Gi Complex Structures."
2. Page 3, line 8: "the one amino acid shift" → "a one-amino acid shift"
3. Page 4, line 18: The sentence is hard to understand, it could be worded as "Despite their structural similarities, the endogenous ligands of GPR119 and CB1 selectively recognize their own receptors".
4. Page 5, line 2: "a hamster β -cell line".
5. Page 5, line 6: The abbreviation "T2DM" needs to be defined.
6. Page 5, line 10: This is the first time that the abbreviation "cryo-EM" is used in the main text, so it needs to be defined.
7. Page 5, line 12: Remove colon after "The structures".
8. Page 6, line 4: Remove colon after "AR231453-".
9. Page 6, line 11: This sentence would be clearer if it indicated what it is being compared. For example: "The root-mean square deviation of the C α s of the two receptors is 0.57 Å".
10. Page 6, line 13: Add a colon after MBX-2982.
11. Page 6, line 20-21: The sentence needs to be reworded. Perhaps: "However, three unusual features distinguish GPR119 from other class A GPCRs".
12. Page 7, lines 1 and 11: "the GPR119 structures".
13. Page 7, line 20: "revealed a one-amino acid shift".
14. Page 8, line 7-10: the sentence is confusing. Perhaps "[...]GPCRs, as comparison to" could be reworded as "[...]GPCRs and in comparison to" or "[...]GPCRs, as shown in comparison to that of b2AR".
15. Page 8, line 12: Instead of "no longer exists", which implies that it existed at some point, I would word this as: "is not present".
16. Page 16, lines 10 and 12: Use 1-letter codes for amino acids to keep the notation consistent.
17. Page 17, lines 15-16: "in other class A GPCR-Gas complexes".
18. Page 19, line 2: Use lower case letters to indicate the figure panels to keep the notation consistent.
19. Page 19, line 7: "majority" → "the majority" or "most"
20. Page 19, line 7: "activation cavity" → "the activation cavity"

21. Page 20, line 17: "AR231453-bound GPR119 structure" → "the AR231453-bound GPR119 structure"
22. Page 21, lines 21-22: "in two agonist-bound systems" → "in the two agonist-bound systems".
23. Page 22, line 2-3: "AR231453 adopted more stable binding poses relative to MBX-2982".
24. Page 22, line 4: "same" → "the same".
25. Page 22, line 5: "ligand-free" system → "the ligand-free system"
26. Page 22, line 9: "displayed more flexibility" or "were more dynamic" would be clearer than "displayed more dynamic".
27. Page 22, line 9 and page 34, line 22: I suggest that the authors use the word "independent" instead of "parallel", when referring to different simulations.
28. Page 33, line 2: "a phenix.elbow" → "phenix.elbow"
29. Page 33, line 23: "docking study" → "a docking study"
30. Page 34, line 2: "in two cryo-EM structures" → "in the two cryo-EM structures"
31. Page 40, line 13: The rest of the PBD codes are capitalized.
32. Page 43, lines 6-8: The wording is confusing. The sentence would be clearer as: "Ligand-receptor interactions of OEA docked at GPCR119 in the extracellular cavity and stacking gate (h) and in the activation cavity (i)."
33. SI Page 10, lines 2-3: "AS1269574 as an agonist" → "agonist AS1269574". Also the panel letters in the legend need to be bolded.
34. SI Page 12, line 6: "500 ns trajectories" → "500 ns of the trajectories"
35. There are multiple instances of "group A family GPCR" or "class A family" (e.g., page 3, line 10; page 6, line 10-11; page 6, line 17; page 7, line 13). I think that this phrasing is redundant. I would substitute it by just "class A GPCRs" or "rhodopsin-like family".
36. Some instances of MBX-2982 are hyphenated, but some aren't (e.g., page 10, line 7 and page 10, line 11).

E. Typos:

1. Page 5, line 8: "synthesis" → "synthetic"
2. Page 7, line 17: "involved" → "involving"
3. Page 9, line 10: "locates" → "located"
4. Page 12, line 3: "closing" → "close"
5. Page 12, line 12: "more closed" → "closer"
6. Page 14, line 7: "reverse" → "reversed"
7. Page 14, line 11: "closed" → "closed"
8. Page 16, line 3: "adopts" → "adopt"
9. Page 17, line 15: "is" → "are"
10. Page 18, line 6: "form" → "forms"
11. Page 40, line 11: "as stick" → "as sticks" or "in stick representation"
12. SI Page 3, line 5: "2.87Å and 2.33Å" → "2.87 Å and 2.33 Å"
13. SI Page 4, line 9: "involved" → "involving"
14. Page 11, line 2 and SI Page 8, line 5: The "..." be an em dash or appear higher in the line.
15. SI Page 8, line 2: "signaling" should be capitalized.
16. SI Page 11, line 6: "closed" → "close to"

Response to reviewers' specific comments:

Reviewer #1 (Remarks to the Author):

In my initial review (for NSMB), I requested that the authors try and get the unliganded or apo state structure for GPR119. I understand that it has been technically challenging to get it and I have nothing more to ask for experiments. I support publication of the paper in its current form, with some language formatting.

The quality of the structures seems very good, and overall it adds to our knowledge of this orphan GPCR.

We appreciate the reviewer's positive evaluation of our work and its significance. We have addressed all of his/her comments with significant changes to the manuscript.

Minor points:

Page 5 line 8: synthetic, not synthesis.

page 7 line 6: extent, not extend.

page 8, line 10: compared, not comparison.

All have been corrected as suggested.

Reviewer #2 (Remarks to the Author):

In response to the questions raised, the authors basically solved my doubts. The description of the content of manuscript has been modified and the format of figure has been optimized, which make the text more logical and clear. However, some words in the manuscript are not precise. For example,

Page 6, line 5, "for all amino acids": it is better to use "for most amino acids".

This sentence has been corrected as suggested.

Page 8, line 21, "with the deepest depth... thus far": it will be more appropriate to use "deeper...than" for these descriptions.

This sentence has been corrected as suggested.

Other similar parts should be checked if I didn't point out.

Reviewer #3 (Remarks to the Author):

Reply from Reviewer 3

The manuscript has been improved with respect to a biological context of GPR119. However, the authors may consider to update the view here with the more recent literature, for instance doi.org/10.1186/s11658-021-00276-7.

The recent literature has been updated and cited.

The author could also improve the manuscript by providing overview of the two selected compounds and their path as putative lead compounds for GPR119

We would like to clarify that we have provided overview of the two

selected compounds in the introduction part, and have discussed SAR for many compounds (supplemental Supplementary Fig. 6). These would provide a path as putative lead compounds for GPR119.

Moreover, the authors may want to correct a few language/typographic mistakes, for instance:

-page 5: line 5, line 8 and 9.

-page 11: line 2

-page 15: line 16

and other places.

All have been corrected as suggested.

Reviewer #4 (Remarks to the Author):

Qian et al. provide structural context for the active state of human GPR119 in the presence of two synthetic agonists and the receptor's cognate G protein. The authors present two high quality cryo-electron microscopy structures of the receptor and identify structural features that have not been reported for other class A G protein-coupled receptors, including an outward bulge in transmembrane helix 5 and a salt-bridge between intracellular loop 1 and the G α subunit. Other protein-ligand interactions that could be relevant for GPR119 activation and signaling are identified from the structures and then experimentally verified through multiple point mutations. The authors also explore the binding mode of different agonists and their derivatives using a docking study and perform molecular dynamics simulations where they compare the dynamics of the ligand-free vs. the agonist-bound receptor. Overall, I believe that the study is thorough and that the reported findings will be of

great interest for the G protein-coupled receptor community. However, the present manuscript would benefit from some revisions, particularly pertaining to the following points:

We appreciate the reviewer's positive evaluation of our work and its significance. We have addressed all of his/her comments with significant changes to the manuscript.

Observations related to the computational work

1. Page 21, line 14-16: It needs to be clarified what subset of atoms was included in the RMSD calculations that are reported in Figure 12a. The loops of GPCRs tend to be quite flexible and it is customary to exclude them when checking the stability of the system. Most groups report the RMSD of the C α s in the transmembrane domain. It is not clear from the text or the figure if this is what the authors did or not?

We appreciate the reviewer's valuable comments. Supplementary Figure 12a showed the RMSD values of ligand in each simulation, and no protein atoms were included in the calculation. We thought the reviewer's comments were about Supplementary Figure 11a. RMSD values shown in Figure 11a were generated using all atoms of protein. Following the reviewer's suggestion, we have recalculated the RMSD for the C α atoms of all transmembrane helices, including S6-L32(TM1), S40-D64 (TM2), K75-I107(TM3), V121-L141 (TM4), H164-I199 (TM5), A223-A250 (TM6) and L258-

W282 (TM7). Similar to previous results, all systems were equilibrated after the first 400 ns (Figure R1).

As expected, the current RMSD values are much smaller than previous data, no more than 2.5 Å. This is consistent with our conclusion that GPR119 reached stability during the simulation. Thus, we've replaced Figure 11a with the new results below.

Figure R1. α -atom RMSD values of the transmembrane domain versus simulation time in three simulation systems.

2. Page 32, line 24: The authors should provide a more complete description of how the initial structure of GPR119 used for refinement was generated. What templates did they use? What were their percentages of identity vs. GPR119? How many models did they generate with MODELLER and how were they ranked (e.g., based on their DOPE score)?

In the docking studies, we have applied two strategies, rigid docking and flexible docking, for all ligands. As shown in Fig. 3g, 3h and 3i, OEA is long and thin, which means it has a smaller volume and could exhibit more conformational flexibility compared to AR231453 or MBX- 2982. Therefore, rigid docking is already enough to obtain good, comparable binding poses for OEA. To better compare OEA's binding mode with AR231453 or MBX- 2982, we chose to use the rigid docking results, for surrounding residues display identical conformations in these complexes. For the other derivatives, most of them has additional moieties compared with AR231453 or MBX- 2982 (Supplementary Fig. 6). Their larger molecular volume causes steric hindrance in the binding pocket and only flexible docking could produce reasonable, comparable binding poses. Though very individual derivatives with smaller volume could also obtain binding conformations under rigid docking, to maintain the consistency in comparison, only flexible docking results were shown for each derivative in Supplementary Figure 7.

According to the above description, we've included corresponding explanations in the method section.

3. Page 34, line 5: Can the authors elaborate on the need to use rigid docking for OEA, but not for the other derivatives that were docked?

In the docking studies, we have applied two strategies, rigid docking and flexible docking, for all ligands. As shown in Fig. 3g, 3h and 3i, OEA is long and thin, which means it has a smaller volume and could exhibit more conformational flexibility compared to AR231453 or MBX- 2982.

Therefore, rigid docking is already enough to obtain good, comparable binding poses for OEA. To better compare OEA's binding mode with AR231453 or MBX- 2982, we chose to use the rigid docking results, for surrounding residues display identical conformations in these complexes. For the other derivatives, most of them has additional moieties compared with AR231453 or MBX- 2982 (Supplementary Fig. 6). Their larger molecular volume causes steric hindrance in the binding pocket and only flexible docking could produce reasonable, comparable binding poses. Though very individual derivatives with smaller volume could also obtain binding conformations under rigid docking, to maintain the consistency in comparison, only flexible docking results were shown for each derivative in Supplementary Figure 7.

According to the above description, we've included corresponding explanations in the method section.

4. Page 34, line 5: Please provide the docking scores obtained from the docking studies.

Docking scores generated from the docking studies are listed below. We didn't provide this table in the manuscript because there is no correlation between docking scores and EC50 of ligands, which may make the readers confusing. This is understandable because the scoring of the docking conformation by Autodock Vina is based on the semi-empirical free energy scoring function, which has some limitations in different situations, and higher scores only indicate better binding interactions between the compounds and receptors in specific computational models but have no direct correlation with experimental activity.

Therefore, the most reliable (or representative) binding poses shown in figures were selected according to both the docking scores and their frequency of occurrence in all obtained conformations for each ligand. For example, if the top-scoring pose only appears once, it will not be considered as a reliable binding pose.

In the tables below, we listed the docking score of the selected pose, which is usually not the one with the best score, as well as the range of the docking score for all conformations. It's obvious that docking scores have no relevance to their efficacy from any perspective. To avoid making readers confusing, we'd better not present these scores in the manuscript.

Table R1. Docking scores of OEA in Figure 3.

Molecule	EC50(μ M) ^a	Docking score (kcal/mol)	Highest score (kcal/mol)	Lowest score (kcal/mol)
OEA	46.84	-7.2	-7.6	-6.6

^aEC₅₀ value of OEA was determined in present work.

Table R2. Docking scores of each derivative in the docking studies. Derivatives are listed by their order in Supplementary Figure 6.

Molecule	EC50(nM) ^a	Docking score (kcal/mol)	Highest score (kcal/mol)	Lowest score (kcal/mol)
Supplementary Fig 6b	3.4	-7.0	-7.8	-5.8
Supplementary Fig 6c	110	-7.5	-7.9	-6.0
Supplementary Fig 6d	7	-7.7	-8.2	-6.1
Supplementary Fig 6e	260	-7.3	-8.0	-6.0
Supplementary Fig 6f	10	-7.4	-7.7	-5.4
Supplementary Fig 6g	1.5	-6.8	-7.7	-6.0
Supplementary Fig 6i	182	-9.4	-10.0	-8.1
Supplementary Fig 6j	49	-10.0	-11.3	-3.6
Supplementary Fig 6k	100	-10.6	-10.6	-8.2

^aEC₅₀ values of derivatives were obtained from previous studies.

5. Page 34, line 20: Do the authors mean “adapted” or “adopted”? Were there penalties higher than 10 in the parameters generated from CGenFF? If so, were the parameters optimized as suggested by Vanommeslaeghe et al. (2012) in their J. Chem. Inf Model 52(12), 3144-54 paper?

We thank the reviewer for the insightful comment. We’ve changed “adapted” to “adopted” in the corresponding text. With regard to the ligand parameters, there are penalties higher than 10, but unfortunately, we didn’t perform optimization at that time. The parameters assigned by CGenFF are listed in detail in the Ligand Parameters section in the Supporting material. Your suggestion is undoubtedly the best in theory, but it needs us to rerun the simulation with optimized parameters, which is usually very expensive in terms of computational resources and time costs. Due to the huge amount of simulation time and computational resources required, we couldn’t provide the updated data in the current response.

Furthermore, we invite the reviewer to consider our following discussion before doing it:

- (1) The purpose of MD simulation in this work is to make more adequate structural comparison between agonist-bound and the predicted ligand-free GPR119 besides static structures. Especially for the predicted AlphaFold structure, MD simulation could optimize its conformation. By comparing conformations of the transmembrane domain as well as ligand binding pocket both from simulation and from static structure, we would be able to draw more convincing conclusions about structural features related to agonist bound. Given that the comparison among static structures already exists (Supplementary Fig. 10), simulation data only plays a supporting role.**
- (2) The structural comparison from MD simulations agreed well with static structures and the accuracy of ligand parameter seems to have little effect on current result. This is reasonable for conformational changes in both the transmembrane domain and the binding pocket. For the**

transmembrane domain, its conformation largely depends on the existence of G protein and it is always difficult to observe significant structural changes of GPCR system within limited simulation time. For ligand surrounding residues, the observed rearrangements in our work were primarily caused by ligand's occupation of pocket space but not brought about by specific polar interactions with the ligand. Therefore, we speculate that MD simulation with optimized ligand parameters would most likely get the same result.

(3) Considering the nonessential role of MD simulation in this work and the little effect of ligand parameter on our conclusion, though new simulations with more accurate ligand parameters would absolutely provide more precise results, it would have no contribution to the innovation and significance of the current manuscript.

6. SI Page 12, line 4: How were the average/representative structures presented in Supplementary Figures 11b, 11d, 12c, 12e and 12f generated? Since it is not really meaningful to average coordinates, usually one would cluster the conformations sampled in the simulations and use the medoid from the most populated cluster as a representative structure.

We thank the reviewer for his/her insightful comment. In Supplementary Figure 11b and 11d, average structures of the whole protein were used to show conformational changes of GPR119. These structures were generated from the last 500 ns in each simulation, for the protein reached relatively stable conformations after 400 ns.

While in Supplementary Figure 12, representative structures of each simulation were presented. In this figure we tried to illustrate the dynamic of bound agonists and their surrounding residues, so average structures of the whole protein in Supplementary Figure 11 were not suitable here. As the movements of bound ligands are highly associated with that of surrounding residues, representative structures were determined based on the ligand RMSD shown in Supplementary Figure 12a and 12d. In the last 200 ns of simulations AR-1, AR-2, AR-3, MBX-1 and MBX-2, fluctuations of RMSD values were small, which meant ligands adopted very stable conformations and in theory every snapshot could equally represent the dynamic features in each simulation. For simulation MBX-3, the ligand exhibited larger fluctuations of RMSD in the last 200 ns, implying it obtained several conformations (Supplementary Fig. 12d). In this case, snapshots in the dominant conformation presenting similar RMSD values could be considered as representative structures, which actually works something like conformation clustering. Based on the above reasons, the snapshot at 900 ns was suitable to be a typical structure for each simulation, so we finally extracted them as representative structures and illustrated their dynamic features in Supplementary Figure 12b, 12c, 12e and 12f.

Following the reviewer's suggestions, we conducted conformation

clustering to show more precise results in modified Supplementary Figure 11 and 12. Protein conformations were clustered based on the RMSD values of transmembrane domain (shown in Supplementary Fig. 11a) using the last 200 ns snapshots in each simulation. The middle structure from the largest cluster was chosen as the representative structure and used for the superimposition in Supplementary Figure 11b and 11d. To present typical ligand binding poses and their surrounding residues, we clustered ligand conformations sampled in the last 200 ns using the ligand RMSD (shown in Supplementary Fig. 12a and 12d). Typical structures were determined in the same way. They were used to illustrate ligand binding conformations in Supplementary Figure 12c and 12f and to present pocket residues in Supplementary Figure 12b and 12e. The updated Supplementary Figure 11 and 12 are displayed below and they reveal similar conformational features to previous figures.

Figure R2. Updated Supplementary Figure 11 containing modified panels a, b and d.

Figure R3. Updated Supplementary Figure 12 containing modified panels b, c, e and f.

Other observations

A. Missing references

1. Page 7, line 5: The reference to the Ballesteros-Weinstein paper needs to be added.

The reference has been added.

B. Data presentation:

1. Page 6, line 12-14: Figure 1 does not really show the two ligands overlapping, but in different panels. And would the fact that they overlap demonstrate that the activation mechanism is conserved? I think that this sentence needs to be reworded to reflect what it is actually in the figure.

We have reworded this sentence to reflect what it is actually in the figure.

2. Figure 2e and f: The different cavities of the ligand binding pocket need to be more clearly

defined, because their limits are not clear just based on the arrows shown in the panels. Perhaps it would be helpful if they were indicated with boxes instead?

We have changed the arrows with circles to clearly define the different cavities.

3. Figure 3a-i: Adding BW-numberings to the residues would be helpful for the reader and it would also match the notation used in the bar plots.

BW-numberings in Figure 3a-I have been added.

4. Figure 4: Since this figure includes mutations in both the receptor and the G protein, I would suggest adding a prefix so that they are easier to identify (e.g., R_ for receptor residues, G_ for G residues and so on).

We would like to clarify that we made all mutations on the receptor.

5. Supplementary Table 1: The value of the rotamer outliers for GPR119-MBX-Gs-Nb35 needs to be centered.

The value of the rotamer outliers has been centered.

C. Technical clarifications:

1. Page 6, line 11 and page 20, line 7: What does the subset of 279 C α s correspond to? The transmembrane helices? It needs to be specified.

We would like to clarify the subset includes 279 residues, the majority of the receptor. We have modified the text.

2. Page 12, line 17: How can ECL2 be part of the activation cavity? Is it not part of the extracellular cavity? Please see my suggestion above of presenting the different cavities in a clearer manner in Figure 2e and f.

We would like to clarify the ECL2 is involved in both the activation cavity and the extracellular cavity.

3. Page 16, line 7-9: I was not able to find the results of the E261A mutant in Figure 3I.

We added the results of the E261A mutant in Figure 3I.

D. Grammar/Phrasing:

1. I would suggest that the authors choose a different title, as the current one is practically identical to Hua et al.'s (2020): "Activation and Signaling Mechanism Revealed by Cannabinoid Receptor-Gi Complex Structures."

We would like to clarify that our structures are GPR119-Gs complex structures, while those of Hua et al.'s are Cannabinoid Receptor-Gi Complex structures.

2. Page 3, line 8: "the one amino acid shift" → "a one-amino acid shift"

3. Page 4, line 18: The sentence is hard to understand, it could be worded as "Despite their structural similarities, the endogenous ligands of GPR119 and CB1 selectively recognize their

own receptors”.

4. Page 5, line 2: “a hamster β -cell line”.

All sentences (2-4) have been corrected as suggested.

5. Page 5, line 6: The abbreviation “T2DM” needs to be defined.

The T2DM have been defined.

6. Page 5, line 10: This is the first time that the abbreviation “cryo-EM” is used in the main text, so it needs to be defined.

The Cryo-EM have been defined.

7. Page 5, line 12: Remove colon after “The structures”.

8. Page 6, line 4: Remove colon after “AR231453”.

Both colons (7-8) have been removed.

9. Page 6, line 11: This sentence would be clearer if it indicated what it is being compared. For example: “The root-mean square deviation of the Calphas of the two receptors is 0.57 Å”.

This sentence has been rephased.

10. Page 6, line 13: Add a colon after MBX-2982.

The colon has been added.

11. Page 6, line 20-21: The sentence needs to be reworded. Perhaps: "However, three unusual features distinguish GPR119 from other class A GPCRs”.

This sentence has been corrected as suggested.

12. Page 7, lines 1 and 11: “the GPR119 structures”.

Both have been corrected as suggested.

13. Page 7, line 20: “revealed a one-amino acid shift”.

This sentence has been corrected as suggested.

14. Page 8, line 7-10: the sentence is confusing. Perhaps “[...]GPCRs, as comparison to” could be reworded as “[...]GPCRs and in comparison to” or “[...]GPCRs, as shown in comparison to that of b2AR”.

15. Page 8, line 12: Instead of "no longer exists", which implies that it existed at some point, I would word this as: "is not present".

Both (14-15) have been corrected as suggested.

16. Page 16, lines 10 and 12: Use 1-letter codes for amino acids to keep the notation consistent.

We have changed the 3-letter codes with 1-letter codes for amino acids.

17. Page 17, lines 15-16: “in other class A GPCR-Gas complexes”.

18. Page 19, line 2: Use lower case letters to indicate the figure panels to keep the notation

consistent.

19. Page 19, line 7: “majority” → “the majority” or “most”

20. Page 19, line 7: “activation cavity” → “the activation cavity”

21. Page 20, line 17: “AR231453-bound GPR119 structure” → “the AR231453-bound GPR119 structure”

22. Page 21, lines 21-22: “in two agonist-bound systems” → “in the two agonist-bound systems”.

23. Page 22, line 2-3: “AR231453 adopted more stable binding poses relative to MBX-2982”.

24. Page 22, line 4: “same” → “the same”.

25. Page 22, line 5: “ligand-free” system → “the ligand-free system”

26. Page 22, line 9: “displayed more flexibility” or “were more dynamic” would be clearer than “displayed more dynamic”.

All (17-26) have been corrected as suggested.

27. Page 22, line 9 and page 34, line 22: I suggest that the authors use the word “independent” instead of “parallel”, when referring to different simulations.

We have used “independent” instead of “parallel”

28. Page 33, line 2: “a phenix.elbow” → “phenix.elbow”

29. Page 33, line 23: “docking study” → “a docking study”

30. Page 34, line 2: “in two cryo-EM structures” → “in the two cryo-EM structures”

All (28-30) have been corrected as suggested.

31. Page 40, line 13: The rest of the PDB codes are capitalized.

All the PDB codes are capitalized as suggested.

32. Page 43, lines 6-8: The wording is confusing. The sentence would be clearer as: “Ligand-receptor interactions of OEA docked at GPCR119 in the extracellular cavity and stacking gate (h) and in the activation cavity (i).”

33. SI Page 10, lines 2-3: “AS1269574 as an agonist” → “agonist AS1269574”. Also the panel letters in the legend need to be bolded.

34. SI Page 12, line 6: “500 ns trajectories” → “500 ns of the trajectories”

All (32-34) have been corrected as suggested.

35. There are multiple instances of “group A family GPCR” or “class A family” (e.g., page 3, line 10; page 6, line 10-11; page 6, line 17; page 7, line 13). I think that this phrasing is redundant. I would substitute it by just “class A GPCRs” or “rhodopsin-like family”.

All “group A” have been substituted with “class A”

36. Some instances of MBX-2982 are hyphenated, but some aren't (e.g., page 10, line 7 and page 10, line 11).

All are hyphenated as suggested.

E. Typos:

1. Page 5, line 8: “synthesis” → “synthetic”

2. Page 7, line 17: “involved” → “involving”
3. Page 9, line 10: “locates” → “located”
4. Page 12, line 3: “closing” → “close”
5. Page 12, line 12: “more closed” → “closer”
6. Page 14, line 7: “reverse” → “reversed”
7. Page 14, line 11: “closed” → “closed”
8. Page 16, line 3: “adopts” → “adopt”
9. Page 17, line 15: “is” → “are”
10. Page 18, line 6: “form” → “forms”
11. Page 40, line 11: “as stick” → “as sticks” or “in stick representation”
12. SI Page 3, line 5: “2.87Å and 2.33Å” → “2.87 Å and 2.33 Å”
13. SI Page 4, line 9: “involved” → “involving”
14. Page 11, line 2 and SI Page 8, line 5: The “...” be an em dash or appear higher in the line.
15. SI Page 8, line 2: “signaling” should be capitalized.
16. SI Page 11, line 6: “closed” → “close to”

All (1-16) have been corrected as suggested

Reviewers' Comments:

Reviewer #4:

Remarks to the Author:

In this second round of revisions, the authors have addressed my concerns thoroughly and made appropriate changes where needed in the manuscript. Particularly, regarding the use of rigid vs. flexible docking protocols for the docking study and the calculation of the RMSDs/representative structures shown in Supplementary Figures 11 and 12. I am content with their reasoning concerning the ligand parameters used in the molecular dynamics simulations, as well. The revised version of the manuscript also incorporated suggestions regarding flow, grammar, phrasing, data presentation and citations in a satisfactory manner. Overall, the text and figures have been significantly improved and I believe that the manuscript is ready for publication. I only have a few minor observations that do not require another round of revisions:

1. Page 6, line 14, 15, 21; page 7, line 16; Supplementary Figure 3, line 5 and 8; page 8, line 13 and 14; Figure 2, line 6; page 22, line 18: remove "family" after "class A"
2. Figure 2, line 8; Supplementary Figure 3, line 11: linking → linked
3. Figure 2, line 7; Supplementary Figure 3: line 8: Please correct the PDB ID of the β 2AR structure. It should be 3SN6.
4. Page 9, line 14: "located" → "is located"
5. Page 10, line 19: "closed" → "close"
6. Page 12, line 7: "close" → "is close"
7. Figure 3, line 18: There is an extra ")" after "tested".
8. Page 17, line 19: "complex" → "complexes"
9. Page 21, lines 4 and 5; page 22, line 9: change the amino acid names to one-letter codes, so that they matched the notation used in the rest of the text and the figures.
10. Page 33, line 1: "Docking-grid" → "Docking grids"

Response to reviewers' specific comments:

Reviewer #4 (Remarks to the Author):

In this second round of revisions, the authors have addressed my concerns thoroughly and made appropriate changes where needed in the manuscript. Particularly, regarding the use of rigid vs. flexible docking protocols for the docking study and the calculation of the RMSDs/representative structures shown in Supplementary Figures 11 and 12. I am content with their reasoning concerning the ligand parameters used in the molecular dynamics simulations, as well. The revised version of the manuscript also incorporated suggestions regarding flow, grammar, phrasing, data presentation and citations in a satisfactory manner. Overall, the text and figures have been significantly improved and I believe that the manuscript is ready for publication. I only have a few minor observations that do not require another round of revisions:

We appreciate the reviewer's positive evaluation of our work and its significance. We have addressed all of his/her comments with significant changes to the manuscript.

1. Page 6, line 14, 15, 21; page 7, line 16; Supplementary Figure 3, line 5 and 8; page 8, line 13 and 14; Figure 2, line 6; page 22, line 18: remove "family" after "class A"
2. Figure 2, line 8; Supplementary Figure 3, line 11: linking → linked
3. Figure 2, line 7; Supplementary Figure 3: line 8: Please correct the PDB ID of the β 2AR structure. It should be 3SN6.
4. Page 9, line 14: "located" → "is located"
5. Page 10, line 19: "closed" → "close"
6. Page 12, line 7: "close" → "is close"
7. Figure 3, line 18: There is an extra ")" after "tested".
8. Page 17, line 19: "complex" → "complexes"
9. Page 21, lines 4 and 5; page 22, line 9: change the amino acid names to one-letter codes, so that they matched the notation used in the rest of the text and the figures.
10. Page 33, line 1: "Docking-grid" → "Docking grids"

All have been corrected as suggested.